# Demographic Evaluation of the Control Potential of *Orius minutus* (Hemiptera: Anthocoridae) Preying on *Dendrothrips minowai Priesner* (Thysanoptera: Thripidae) at Different Temperatures

**DOI:** 10.3390/insects13121158

**Published:** 2022-12-15

**Authors:** Rongmeng Lan, Xiaoli Ren, Kunqian Cao, Xia Zhou, Linhong Jin

**Affiliations:** 1State Key Laboratory Breeding Base of Green Pesticide and Agricultural Bioengineering, Key Laboratory of Green Pesticide and Agricultural Bioengineering, Ministry of Education, Guizhou University, Huaxi District, Guiyang 550025, China; 2College of Agriculture, Guizhou University, Guiyang 550025, China

**Keywords:** *Orius minutus*, tea thrips (*D. minowai*), age-stage, two-sex life table, predation rate, temperature, population prediction

## Abstract

**Simple Summary:**

*Orius minutus* (Hemiptera: Anthocoridae) has been used as an effective biological control agent (BCA) for many thrips pest in different crops. However, its predating ability on tea thrips, a major pest in tea garden, has not yet been discerned. Meanwhile the effect of temperature on their biology and interaction is unknown. We here first evaluated the population development of this predator interaction with pests by using age-stage, two-sex life tables analysis under a range of constant environmental conditions. We proved that the minute pirate bug could effectively control the population development of tea thrips under intermediate temperatures (20–25 °C). Both predator and pest were extremely vulnerable at 35 °C. The present result might be used to predicate the population dynamics and downstream applications for *O. minutus* as BCA against tea thrips in tea plantation.

**Abstract:**

Tea thrips (*Dendrothrips minowai* Priesner) are the main pests that seriously affect the yield and quality of tea, resulting in huge economic losses. The *Orius minutus* is one of the most important natural enemies or BCA of thrips. However, we are not concerned with its predation ability on tea thrips, nor thermal influence on this pattern and their interaction. Therefore, this study recorded life table data of *O. minutus* and tea thrips combined with predation rate data to assess the ability of *O. minutus* to control tea thrips using age-stage, two-sex life tables at five constant temperatures. The results showed that at 25 °C, *O. minutus* had the highest predation rate on tea thrips, with an average generation time (*T*) of 22 d, intrinsic rate of increase (*r*) of 0.12 d-1, fecundity of 64.17, net reproduction rate (*R*_0_) of 12.76 offspring, and net predation rate (*c*_0_) of 310.92. In addition, around 410,000 adults and 1.98 million eggs were produced within 120 days. While the temperature change was straightforward, temperature effects on insects are not linear. The population size of the *O. minutus* and tea thrip trended similarly at 15–30 °C and would eliminate dramatically at 35 °C. Meanwhile, the results indicated that *O. minutus* could effectively inhibit the population growth of tea thrips at 15–30 °C, within 5–19 days at an intervention ratio of 10 adult *O. minutus* and 200 thrips individuals. The simulations under different mediated temperatures demonstrated that *O. minutus* is effective against tea thrips over a wide temperature range expected to be potential for biocontrol of tea thrips in tea gardens.

## 1. Introduction

Tea thrips (*Dendrothrips minowai* Priesner) (Thysanoptera: Thripidae) are common throughout the tea garden in the world and caused serious damage to tea yield and quality and economic losses each year [1,2]. At present, the control of these tea pests has been highly dependent on the frequently applied broad-spectrum insecticides. However, thrips, due to their small size, can better avoid pesticides by hiding and feeding in the buds and crevices of tea trees [3]. At the same time, the frequent use of chemical pesticides can cause irreparable damage to the ecological environment of tea plantations. Biological control is one of the more effective methods to control the pest tea thrips in terms of organic agriculture and environmental biodiversity.

Minute pirate bugs, a genus of predatory anthocorids insect in the order Hemiptera, are considered beneficial as they feed on small pest arthropods and their eggs, such as thrips, whiteflies, mites, leafhoppers, and other small pests [4,5,6,7]. Among them, *O. minutus* was first reported as a predatory natural enemy for the control of three species of thrips (Anthophilous: Thysanoptera) [8]. *O. minutus* in Korea has been developed as a biological control agent (BCA) against T. urticae [9] and could be mass-reared at low-cost [10]. *O. minutus* was also applied to the biological control of *Odontothrips loti* (Hal.) [11]. Though *O. minutus* was often found in tea garden, its biological control potential on tea thrips (*D. minowai*), a main teat pest in tea plant [12,13], was unknown.

In addition, temperature has the potential to affect every aspect of an insect’s biology and development [14,15,16]. Understanding the effects of temperature on predators and prey is particularly important given the thermal change during the future field application. However, thermal influence on *O. minutus* and their interaction with thrips was not reported. Previous reports about anthocorid predators are limited to the temperature influence on natural enemies and ignore or detached the temperature effect on pest and pattern interaction. For example, *Orius strigicollis* as a BCA has been reported for the thermal influence on the population growth without considering the simultaneous alert on its prey *Pectinophora gossypiella* at three constant temperatures (24, 28, and 32 °C) [17]. Growth and development of *Orius majusculuss* were evaluated under nine constant temperature conditions ranging from 12 to 34 °C [18].

The age-stage, two-sex life table is the main tool for exploring the characteristics of insect populations by assessing the age structure, survival rate, developmental rate, and fecundity of the population, and sex and stage differentiation [19]. Compared to the traditional life table [20], the age-stage, two-sex life table is able to evaluate the demographic development of natural enemy insects and provide a complete description of the biological control potential of this species [21,22,23,24]. The application of this program and theory has made significant promotion in the research of population ecology, biological control and pest management and was widely accepted by many entomologists [25,26,27].

To evaluate the control potential of *O. minutus* against the tea thrips (*D. minowai)*, we applied the age-stage, two-sex life table to analyze the life table data and predation rate; furthermore, we used computer simulation to predict the growth of the predator population and the change of predation rate. The results provided a theoretical basis for the prevention and control of *D. minowai*.

## 2. Materials and Methods

### 2.1. Rearing of Insects

*D. minowai* and *O. minutus* were collected from a tea gardern in Meitan County (27°20′18″ N, 107°15′36″ E). *D. minowai* was reared for more than ten generations on potted tea plants in a greenhouse at a temperature of 22 ± 1 °C, relative humidity (RH) of 65 ± 5%, and a natural photoperiod of 16 h: 8 h (L:D). In addition, *O. minutus* was reared for more than ten generations on potted Kidney bean pods (*Phaseolus vulgaris* L.), which were used as an oviposition substrate for *O. minutus* and to maintain proper humidity in an insect box (20 cm × 25 cm × 10 cm). The cage was enclosed by a 100-mesh nylon net with a zipper. *D. minowai* were daily collected from tea plants with a fine brush and provided to *O. minutus*. The rearing procedure was carried out in artificial climate incubators at a temperature of 25 ± 1 °C, relative humidity (RH) of 65 ± 5%, and a natural photoperiod of 16 h:8 h (L:D).

### 2.2. Life Table and Predation Rate Studies at Different Temperatures

#### 2.2.1. Life Table and Predation Rate Study of *O. minutus*

One hundred eggs of *O. minutus* laid within 24 h were collected for life table and predation rate studies at 15, 20, 25, 30, and 35 °C. Each newly emerged nymph was transferred to a rearing box (20 cm × 25 cm × 10 cm). Because all stages of *O. minutus* can kill 2nd instar *D. minowai*, 2nd instar thrips were used as prey. Developmental duration and survival were observed daily. Newly emerged adults were paired, and each pair was kept in a box. One hundred 2nd instars of *D. minowai* were supplied to paired *O. minutus* daily. Kidney beans pods from the previous day were examined under a stereomicroscope to record the daily the number of eggs laid by *O. minutus* and transferred to a new cage. The kidney bean pods were replaced daily to record fecundity and predation rate of *O. minutus*. New recruits of *D. minowai* were supplied to maintain a constant number of prey (about 100 of 2nd instars of *D. minowai*). The killed prey was distinctively shrunken and thrips that died naturally stayed in their natural distended shape. Thus, the prey that died due to predation could be easily identified.

#### 2.2.2. Life Table Study of *D. minowai*

Five insect cages with around 200 newborn eggs were put, respectively, in five different artificial climate chambers of 15, 20, 25, 30, and 35 °C. Developmental duration and survival were observed daily. After the emergence of adult *D. minowai*, the number of eggs newly laid each day was recorded until all the adults died.

### 2.3. Data Analysis

#### 2.3.1. Two-Sex Life Table Analysis

Analysis of life table raw data was performed based on the age-stage, two-sex life table theory [28,29,30] by using the computer program TWOSEX-MSChart (2021-11-20-B100000.exe, Taiwan, China) [31]. The parameters including the age-stage growth rate (*g_xj_*), developmental rate (*d_xj_*), age-stage-specific survival rate (*s_xj_*), age-specific survival rate (*l_x_*), age-stage specific fecundity (*f_xj_*), age-specific fecundity (*m_x_*), net maternity (*l_x_m_x_*), intrinsic rate of increase (*r*), finite rate of increase (*λ*), net reproductive rate (*R*_0_), and mean generation time (*T*) were calculated accordingly referring to the report [29].

The *g_xj_* is the probability that an individual of age *x* and stage *j* will survive to age *x* + 1 but remain in the same stage *j*. The *d_xj_* is the probability that an individual of age *x* and stage *j* will survive to age *x* + 1 and develop to the next stage [25,29]. The age-stage-specific survival rate (*s_xj_*) is the probability that a newborn individual survives to age *x* and stage *j*. The simplified formula for age-stage specific survival is as follows [29].
(1)$$s_{xj}=\frac{n_{xj}}{n_{01}}$$
where *n*_01_ is the number of new individuals used at the start of the life table study and *n_xj_* is the number of individuals that survive to age *x* and stage *j*. The age-specific survival rate (*l_x_*) and the age-specific fecundity (*m_x_*) calculations are as follows.
(2)$$l_{x}=\sum_{j=1}^{\beta }s_{xj}$$
(3)$$m_{x}=\frac{\sum_{j=1}^{\beta }s_{xj}f_{xj}}{\sum_{j=0}^{\beta }s_{xj}}$$
where *β* is the number of stages, and *f_xj_* is the age-stage specific fecundity. The intrinsic rate of increase (*r*) is estimated by the Euler-Lotka formula with the age indexed from 0 [32].
(4)$$\sum_{x=0}^{\infty }e^{-r(x+1)}l_{x}m_{x}=1$$

The finite rate of increase (*λ*) is calculated as
(5)$$\lambda =e^{r}$$

The net reproductive rate (*R*_0_), which is the average number of offspring that an individual can produce during its lifetime, is calculated as follows.
(6)$$\sum_{x=0}^{\infty }l_{x}m_{x}=R_{0}$$

The mean generation time (*T*) refers to the time that the population needs to increase to the stable distribution of *R*_0_ under the stable age distribution. The calculation formula is as follows.
(7)$$T=\frac{\ln R_{0}}{r}$$

The age-stage life expectancy (*e_xj_*) is the time that an individual at age *x* and stage *j* is expected to live after age *x* is calculated as [33]:(8)$$e_{xj}=\sum_{i=x}^{\infty }\sum_{y=j}^{\beta }{s^{\prime }}_{iy}$$
where ${s^{\prime }}_{xj}$ is the probability that an individual of age x and stage j survives to age i and stage y.

The age-stage reproductive value (*v_xj_*) is the reproductive contribution of individuals at age *x* and age *j* to the future population, and the formula is as follows [34].
(9)$$v_{xj}=\frac{e^{r(x+1)}}{s_{xj}}\sum_{i=x}^{\infty }e^{-r(x+1)}\sum_{y=i}^{\beta }{s^{\prime }}_{iy}f_{iy}$$

#### 2.3.2. Predation Rate Analysis

The daily predation rates of *O. minutus* were analyzed with reference to Chi and Yang’s method [35] assisting with computer program CONSUME-MSChart (2021-11-20-B100000.exe, Taiwan, China) [36]. The age-specific predation rate (*k_x_*), the age-stage specific predation rate (*c_xj_*), age-specific net predation rate (*q_x_*), net predation rate (*c*_0_), and transformation rate (*Q_p_*) were calculated as follows.

The age-specific predation rate (*k_x_*) is calculated as follows.
(10)$$k_{x}=\frac{\sum_{j=1}^{\beta }s_{xj}c_{xj}}{\sum_{j=1}^{\beta }s_{xj}}$$
where *c_x_*_j_ is the average number of *D. minowai* consumed by individuals at *x* age and *j* stage, and it is calculated as
(11)$$c_{xj}=\frac{\sum_{i=1}^{n_{xj}}d_{xj,i}}{n_{xj}}$$
where *d_xj,i_* is the predation rate of the *i*-th individual at age *x* and stage *j*.

By taking into consideration of the survival rate, the age-specific net predation rate *q_x_* is calculated as:(12)$$q_{x}=l_{x}k_{x}=\sum_{j=1}^{\beta }s_{xj}c_{xj}$$

The net predation rate (*c*_0_) refers to the average amount of prey consumed by a predator in its lifetime, which includes all individuals that died before the adult stage and those that survived to the adult stage, and is calculated as follows.
(13)$$c_{0}=\sum_{x=0}^{\infty }\sum_{j=1}^{\beta }s_{xj}c_{xj}=\sum_{x=0}^{\infty }l_{x}k_{x}$$

The transformation rate (*Q_p_*) is defined as the amount of prey required by a predator to produce a single offspring and is calculated as follows.
(14)$$Q_{p}=\frac{c_{0}}{R_{0}}$$

Two stage-specific predation rates *P_j_* and *U_j_* are calculated according to Ding et al. [25]. *P_j_* is calculated using only individuals that successfully completed stage *j* and survived to stage *j* + 1, while *U_j_* is calculated using all individuals surviving at the beginning of stage *j*:(15)$$P_{j}=\frac{\sum_{i=1}^{n_{j}}p_{ij}}{n_{j}}$$
(16)$$U_{j}=\frac{\sum_{i=1}^{m_{j}}p_{ij}}{m_{j}}$$
where *n_j_* is the number of individuals that survived from stage *j* to *j* + 1, *m_j_* is the number of individuals that were surviving at the beginning of stage *j*, and *p_ij_* is the total predation rate of individual *i* in stage *j*. Therefore, it is clear, that if there is no mortality in stage *j*, then *m_j_* = *n_j_* and *P_j_* = *U_j_*.

The finite predation rate (*ω*) was calculated according to Yu [30] as:(17)$$\omega =\lambda \psi =\lambda \sum_{x=0}^{\infty }\sum_{j=1}^{\beta }a_{xj}c_{xj}$$
where *a_xj_* is the proportion of individuals belonging to age *x* and stage *j* in a stable age-stage distribution, and *ψ* is the stable predation rate.

#### 2.3.3. Population and Predation Projections

The age-stage growth rate (*g_xj_*), developmental rate (*d_xj_*), and fecundity (*f_xj_*) acquired in the two-sex life table study were applied to simulate the growth of *O. minutus* populations according to Chi [25,37] by using the computer program TIMING-MS Chart (2021-11-20-B100000.exe, Taiwan, China) [38]. The population growths at five temperature treatments were simulated using an initial population of 10 newly emerged eggs.

The formula for calculating the total population size *N*(*t*) at time *t* is as follows.
(18)$$N(t)=\sum_{j=1}^{\beta }\sum_{x=0}^{\infty }n_{xj,t}$$
where n*_xj,t_* is the number of individuals at time *t* and stage *j*. The predation potential at time *t* was calculated as:(19)$$P(t)=\sum_{j=1}^{\beta }\sum_{x=0}^{\infty }c_{xj}n_{xj,t}$$

## 3. Results

This section may be divided by subheadings. It should provide a concise and precise description of the experimental results, their interpretation, as well as the experimental conclusions that can be drawn.

### 3.1. Two-Sex Life Table and Predation Rate

The duration of each stage of the *O. minutus* was significantly affected by temperature (Table 1). For example, the specific stage and total longevity of *O. minutus* reared at low temperatures were increased when compared to higher temperature regimes. The egg duration was 14.42 days at 15 °C but only 1.57 days at 35 °C. The total preadult duration was 41.50 days at 15 °C and shortened to 11.07–11.16 days at 30 °C to 35 °C. There was similar impact on both male and female insects in linear temperature change with no significant difference at the same temperature. The total lifespan of both females and males shortened with the increase in temperature from 15 to 35 °C, with the longest duration of 65.56 days (female) and 59.20 days (male) at 15 °C, and the shortest at 14.17 days (female) and 15.25 days (male) at 35 °C. The longest adult preoviposition period (APOP) was 9.67 days at 15 °C, while the shortest was 1.57 days at 30°C. In addition, the oviposition time was the longest 10 days at 25 °C, and dropped to the shortest 2 days at 35 °C. At the same time, fecundity (*F*) was highest at 25 °C with 64.17 eggs produced per female. There were also great differences in the ratio of males and females at different temperatures and the total adult number and the female number was highest at 25 °C with high potential for reproduction (Table 1 and Table 2).

Likewise, the temperature had a significant effect on the population statistic parameters of the *O. minutus* including mean generation time, net predation rate, *c*_0_ (prey/individual), finite predation rate, *ω* (d^−1^), transformation rate, *Q_p_* (*c*_0_/*R*_0_), gross reproduction rate (*GRR*), and fecundity (number of eggs from per female) (Table 2). As the temperature increased, the mean generation time (*T*) was reduced from 56.27 days at 15 °C to 13.79 days at 35 °C. The intrinsic rate of increase (*r*) was significantly higher at 25 °C than that of other temperatures (35 °C provides an extremely low negative value of −0.1132). The finite rate of increase (*λ*) at 25 °C was 1.1302 d^−1^, significantly higher than that of other temperatures. The net reproductive rate (*R*_0_ = 14.76) was highest at 25 °C and lowest (*R*_0_ = 0.21) at 35 °C. The net predation rate (*c*_0_) differed significantly among the five temperatures and was highest at 25 °C with 310.29 preys and lowest 17.29 preys at 35 °C. The finite predation rate (*ω*) was 9.85 preys per day at 25 °C and 7.41 preys per day at 30 °C and the lowest 2.35 preys per day at 15 °C. The gross reproduction rate (*GRR*) was 42.23 eggs per *O. minutus* at 25 °C (highest) and 2.42 eggs at 35 °C (lowest). The transformation rate (*Q_p_*) was also the most efficient at 25 °C where the *O. minutus* consumed the lowest preys (21.07 individuals of *D. minowai*) to produce one egg (Table 1 and Table 2).

We observed and calculated the age-stage specific survival rate (*s_xj_*) and age-stage specific predation rate (*c_xj_*). of *O. minutus* starting from 100 eggs by feeding on second nymph *D. minowai* at 15–35 °C (Figure 1). As the development time between individuals was different, the age-stage-specific survival rate (*s_xj_*) among several stages overlapped (Figure 1A). Mean longevity of *D. minowai* was significantly affected under each observed constant temperature and the growth time *O. minutus* was linearly shortened from around 73 days to 16 days with increasing temperature from 15 to 35 °C. However, the survival rate was not linearly changed, and performed highest at 25 °C. In addition, the increased age-stage specific predation rate (*c_xj_*) (Figure 1B) were observed during the *O. minutus* growth from the first nymph to the adult stage during each constant temperature, the predation rate was similar among intermediate temperatures from 15 to 30 °C, and the consumption at the adult stage accounted for more than half of its lifetime predation, except at 35 °C (Figure 1B, Table 3).

To compare the predation rates of each developmental stage at different temperatures, we calculated the total predation rate of the stages, *P_j_* (including only individuals that successfully completed stage *j* and survived to stage *j* + 1) (Table 3), and *U_j_* (including individuals surviving at the beginning of stage *j*) for all individuals that survived to enter this developmental stage (Table 4). During the preadult stage, the total predation rate was thought to increase with stage development, while it was revealed higher at the third nymph stage as in tests at 25 °C and 35 °C (Table 3 and Table 4).

In addition, age-specific survival rate (*l_x_*) and age-stage specific fecundity (*f_x_*_7_) were calculated and compared (Figure 2). It is obvious that fecundity *O. minutus* was strongest at 25 °C during egg laying period from day 15 to day 28, with highest 11 eggs at day 22. The age-specific net predation rate (*qx*), age-specific predation rate (*k_x_*), and age-specific survival rate (*l_x_*) indicated 25 °C would be most accommodate for the develop of predators than other test temperatures (Figure 3).

The age-specific fecundity *m_x_* and *l_x_m_x_* and the age-stage life expectancy *e_xj_* for each treatment together demonstrate the overall effect of different temperatures on the *O. minutus* population (Figure 2 and Figure 4A). The adult contribution to future populations was much greater than that of the nymph at the five temperatures, and the female adult contribution to future populations was much higher than that of the male adult and nymph at all ages at 15–25 °C. The male contribution increased in the population class at 30 °C and 35 °C, with a higher peak at 35 °C than the female adult, and the female adult contribution to future populations was greatest at 25 °C. The results showed that the day of the highest reproductive value (*v_xj_*) was always close to the day of total preoviposition (TPOP) [39]. The highest reproduction value occurred on the 18th day at 25 °C, which was close to the day of TPOP (16.87). The highest reproductive value appeared on the 50th day at 15 °C, which was close to the days of TPOP (51.11). The highest reproduction value appeared on 8th day at 35 °C, which was close to the day of TPOP (12.5) (Table 1, Figure 4B).

In bootstrap, random resampling and replacement from the original cohort are used to detect possible recombination individuals in the sample, and then SE is estimated using the mean of all samples (in this research, B = 100,000).

### 3.2. Population Projection

We projected the population growth of *O. minutus* using the achieved life table data while supposing constant and sufficient prey was provided. We tested how variation in temperature affects pea aphid population size

Each of the 10 eggs of *O. minutus* were set as initial subjects at five constant temperatures 15–35 °C, respectively, using the computer program TIMING-MSChart to and evaluated for their dynamics development over a period of 120-day (Figure 5). The projection results illustrated that the population of *O. minutus* expands exponentially to over 1.98 million eggs, four million nymphs, and exceeding 0.41 million adults by day 120 at 25 °C. A slow growth rate with fluctuation at 30 °C was also observed. In addition, the undulant curve at 15 and 20 °C indicated a fluctuating population during those 120 days. At 35 °C, the population declined to almost elimination on day 36. The growth trend was slow and close to a linear growth process in other temperatures including 15, 20, and 30 °C.

*O. minutus* populations were strongly affected by temperature with populations being reduced by more than ten times the size of the populations in the lowest temperature, and the other treatments being in between (Figure 6). This lead to a significant effect of temperature treatment (F(3,88) = 36.09, *p* < 0.0001).

In the meantime, we also studied the life table of *D. minowai* in group rearing under the same five temperature conditions to evaluate the control potential of *O. minutus* on tea thrips. This investigation was initiated using 200 viable eggs of *D. minowai* at 15, 20, 25, 30, and 35 °C, respectively, using the computer program TIMING-MSChart to obtain its developmental dynamics in a period of 120 days (Figure 6). The stage structures of *D. minowai* beginning with an initial population of 200 eggs are shown in Figure 7. Population projections showed a rapid increase in population size at 25 °C and arrived 30 million at 120 days. A similar but quite slower growth rate was also observed at 30 °C, and the fluctuating population at 15 and 20 °C. The *D. minowai* showed a negative population growth at 35 °C, which is consistent with our previous survey result that few tea thrips could be found in the field in high-temperature season.

### 3.3. Population Growth and Predation Capacity

We then simulated stage sizes of *D. minowai* starting form 200 eggs under 15, 20, 25, 30, and 35 °C, with five pairs of adult *O. minutus* introduced on the tenth day (Figure 7). The simulation results showed *O. minutus* can eliminate the *D. minowai* population and effectively control the development of the *D. minowai* population at all temperature ranges. *D. minowai* could not survive to the adult stage at 15 °C and 20 °C and did not produce offspring, and the preadult lasted for 28 day and 18 days, respectively. This is because the *D. minowai* adults took over 10 days to grow from eggs at 15 °C (Figure 5), and once appeared, they were be predated by *O. minutus,* which were introduced at the 10th day. So, there was no adult stage observed in the simulation (Figure 6). At 25 and 30 °C, *D. minowai* might survive to the adult stage, but only lasted for a couple of days and were consumed at the release day. All *D. minowai* would be predated at the 15th and 25th day, about 5–19 days after the release of *O. minutus*.

## 4. Discussion

The age-stage, two-sex life table can accurately describe the demographic variability of insects and their stage differentiation predation rate and adaptive ability [25] and evaluate the biological control efficiency of predators [30,40,41] This theoretical approach has been recognized by more and more entomologists and ecologists.

Here in this study, we applied the age-stage, two-sex life table to describe the population variation and predation ability of *O. minutus* as BCA on *D. minowai*. The simulation of interactions between natural enemies and pests is fundamental to biological control applications [42]. We performed the related experiments by manipulating the exposure temperature (15, 20, 15, 20, 25, 30 °C) for all insects and tested how variation in temperature affects population size *O. minutus*. and *D. minowai* and their interaction. We first investigated the changes in the growth time of *O. minutus* and tea thrips at five constant temperatures. It was found a linear dependence on temperature with increased stage or total longevity of *O. minutus* at lower temperature regimes. Moreover, the population development of both *D. minowai* showed an upward trend under the temperature range of 15–30 °C (Figure 1). In addition, *O. minutus* and tea thrips could not complete generation development at 35 °C. The results suggested that constant high temperatures greatly affected insect behavior, survival, development, reproduction, and other biological characteristics due to the limitation of their ability to regulate body temperature as ectotherms [41,42]. 

The net predation rate (*c*_0_) of *O. minutus* at 25 °C was 310 prey (Table 2), and higher than a commercialized anthocorid BCAs, for example, *O. strigicollis,* which consumed an average of 140 western flower thrips or 101 *Frankliniella intons* during its lifetime at constant 25 °C [25]. This result may imply that *O. minutus* has great potential as BCA on tea thrips [26,27,39]. Meanwhile, we simulated the 120-day population development of the predator and prey, and the interaction between them at five constant temperatures at a ratio of 10 *O. minutus* to 200 prey (Figure 6). We projected that the minute pirate bug could effectively control the population development of tea thrips under intermediate temperatures (15–30 °C), where it might take 5–19 days for thrips to be fully predated. At 35 °C, the population size of *O. minutus* died out before that of tea thrips. According to the previous data, the intrinsic growth rate (*r*) of both of them is less than 0 in the environment of 35 °C and tea thrips will gradually die out in this environment even without the intervention of the natural enemy of the predator *O. minutus*.

We saw that both were in high fitness at 25 °C and vulnerable to the higher temperature range (Figure 5 and Figure 6). Our studies provide strong evidence of increased longevity in BCAs reared at low temperatures when compared to higher temperature regimes which were highly matched with the profile of *Diaphorencyrtus aligarhensis* (Hymenoptera: Encyrtidae) as BCAs in field releases in California [43,44]. They also prove fluctuating temperature profiles produced smaller populations compared to the constant temperatures. As our present simulation was under the assumption of unlimited food, constant temperature, and suitable egg-laying substrate, there would be disagreement with real natural habitats involving more factors, including seasonal temperature fluctuations, searching rate, predators’ conservation, an competitor and target pest development (supply) [45,46]. Meanwhile, in the field environment, the population development of natural enemies and pests is dynamically balanced, and pests could not be eliminated as rated consumption in lab modulating.

More specifically, our simulation suggested that the *O. minutus* can effectively inhibit the population growth of *D. minowai* in a range of constant temperature conditions. The possibility of fluctuating temperature profiles was not investigated in this study. However, the presence of *D. minowai* infestation in field is in naturally fluctuating environmental conditions variating in intensity and duration of temperature. Hence the present reproductive and predation parameters of *O. minutus as* BCAs on *D. minowai* might introduce deviations from the realistic performance. It is by no means simple to compare results across laboratory studies and to extrapolate findings to field conditions [43,44]. Still, studies on thermal biology in ectotherms under constant laboratory conditions are key to obtain a better understanding and prediction of the performance of this pattern of predation.

## 5. Conclusions

We used the age-stage, two-sex life table method and integrated with the predation rate of *O. minutus* feeding on *D. minowai* predicted and proved *O. minutus* was effective in *D. minowai* controlling at five constant temperature ranges: 15, 20, 25, 30, and 35 °C. Our results demonstrated that computer simulations are a reliable tool and the performance of biological control agents *O. minutus* preying on *D. minowai* were achieved at constant temperature, though fluctuating environmental conditions were not included. In fact, we are limited in managing the exact demographic estimation at our current understanding. A further consideration in future field applications would help to certify the population projection and acquire a more exact result and understanding.

In conclusion, this research confirmed that the *O. minutus* possess high biological control potential to the *D. minowai* in a wide temperature range. Incorporation of these results into biological control on *D. minowai* with future field performance may streamline mass-rearing, and improve the accuracy of predictions of *O. minutus*.

## Figures and Tables

**Figure 1 insects-13-01158-f001:**
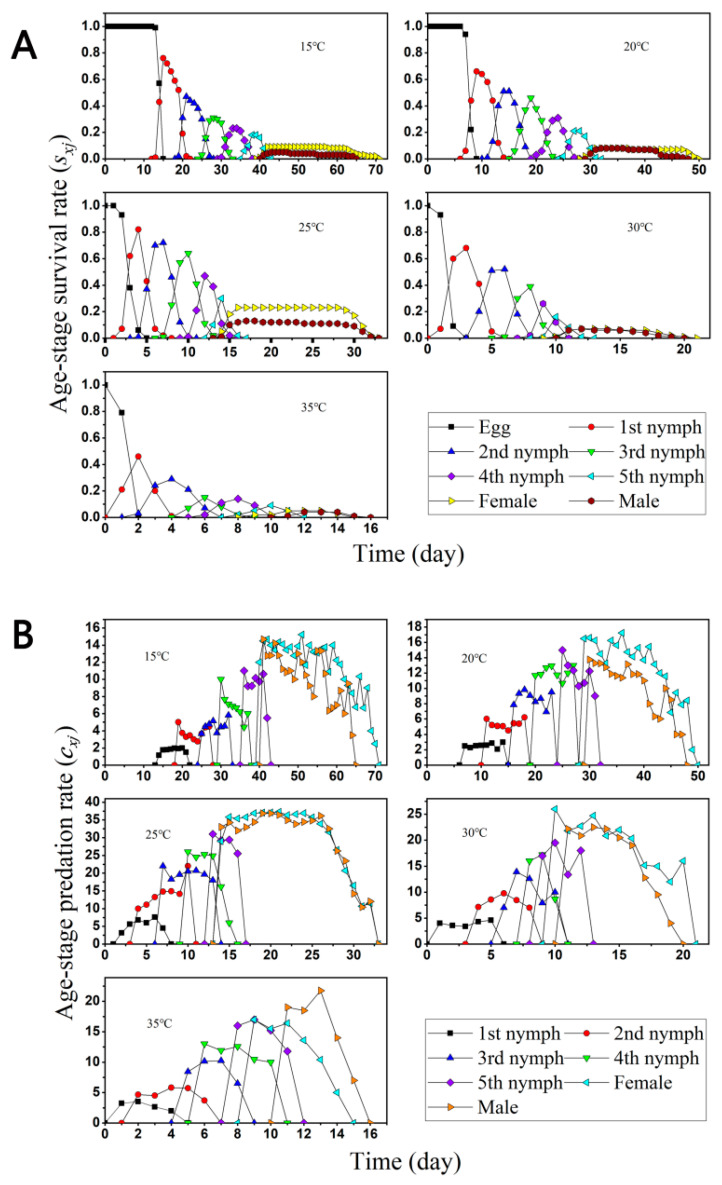
Age-stage specific survival rate (*s_xj_*) (**A**) and age-stage specific predation rate (*c_xj_*). (**B**) of *O. minutus* starting from 100 eggs fed on *D. minowai* at different temperatures.

**Figure 2 insects-13-01158-f002:**
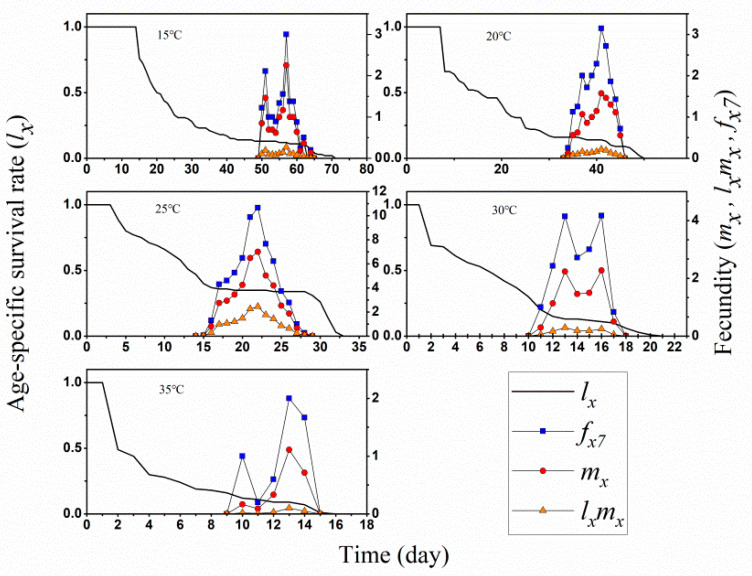
Age-specific survival rate (*l_x_*), age-stage specific fecundity (*f_x_*_7_) (i.e., female adult is the 7th life stage), age-specific fecundity (*m_x_*), age-specific maternity (*l_x_m_x_*) of *O. minutu* at different temperatures.

**Figure 3 insects-13-01158-f003:**
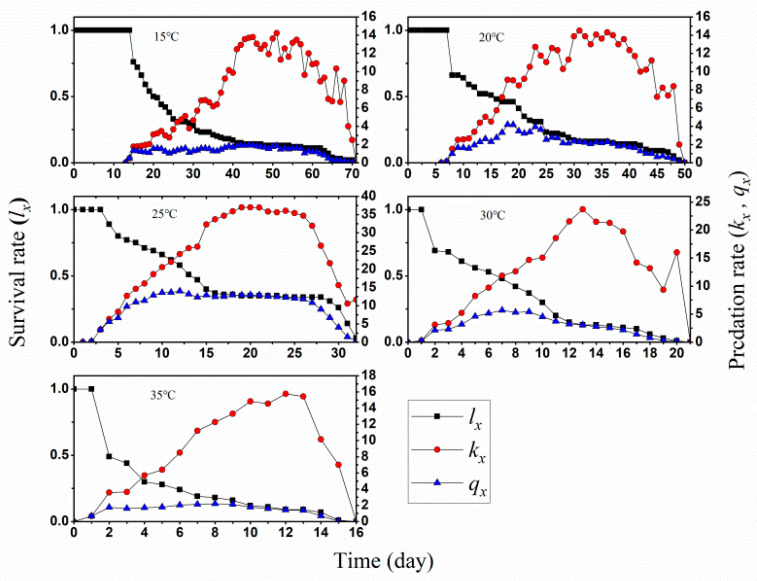
Age-specific survival rate (*l_x_*), age-specific predation rate (*k_x_*) and age-specific net predation rate (*q_x_*) of *Orius minutus* at different temperatures.

**Figure 4 insects-13-01158-f004:**
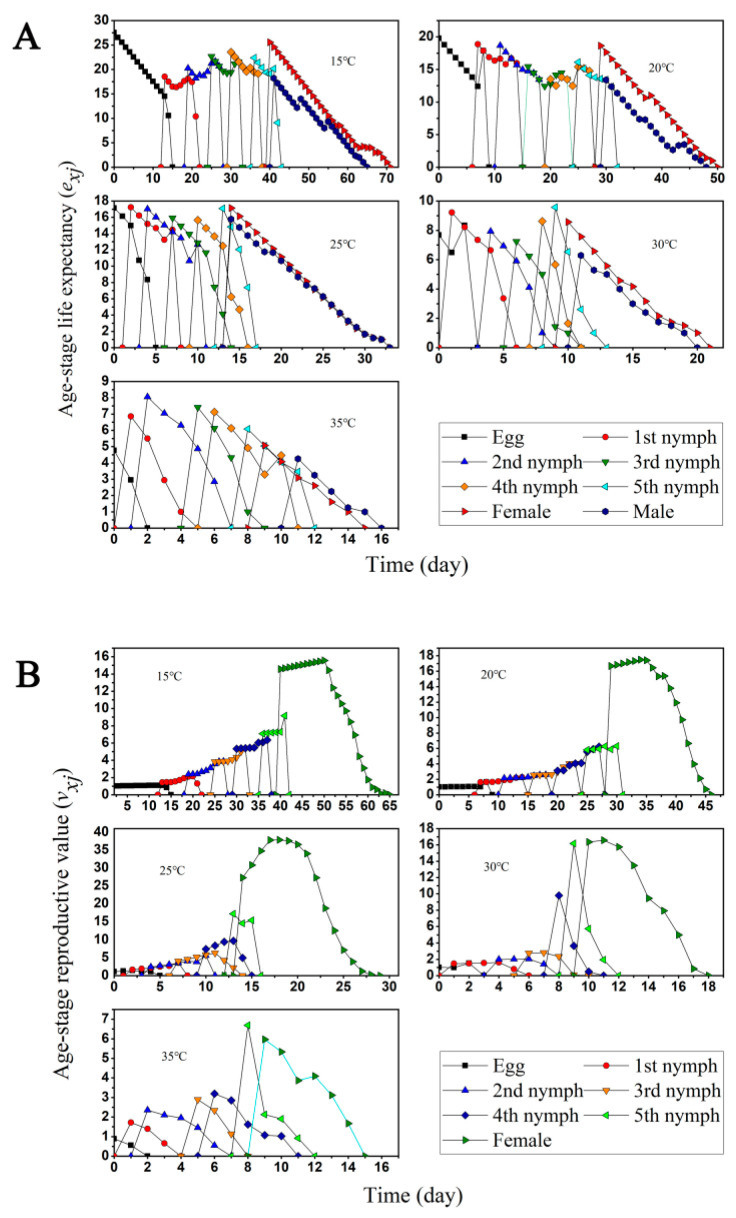
(**A**) Age-specific life expectancy (*e_xj_*) and (**B**) age-stage reproductive value (*v_xj_*) at five different temperatures.

**Figure 5 insects-13-01158-f005:**
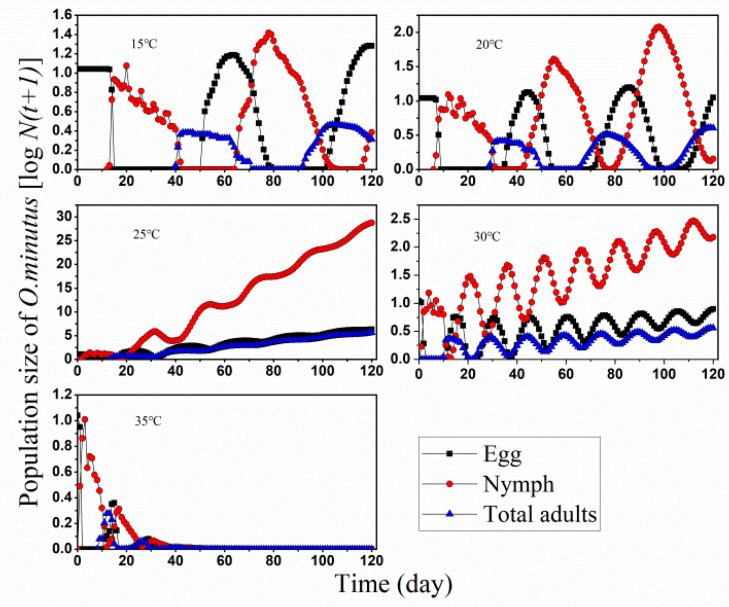
Simulation of *O. minutus* population fed on *D. minowai* starting from 10 eggs at five temperatures from 15 to 35 °C through a period of 120 d.

**Figure 6 insects-13-01158-f006:**
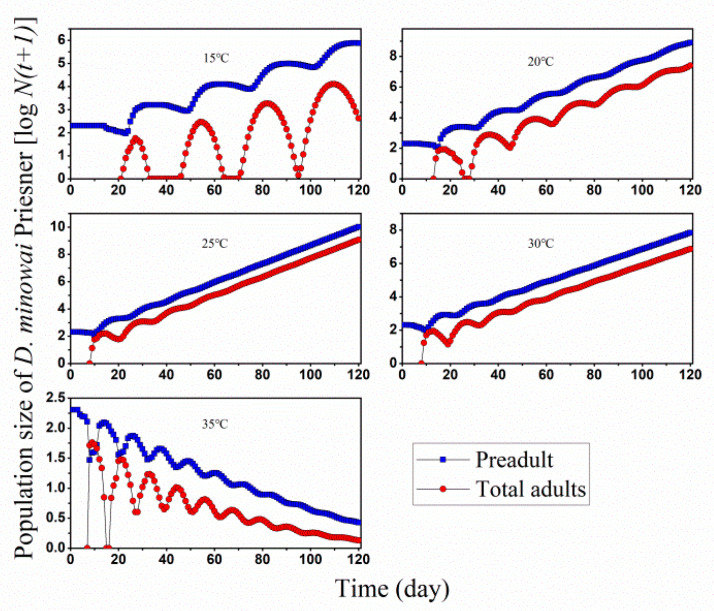
Simulation of *D. minowai* population size starting from 200 eggs at different constant temperatures.

**Figure 7 insects-13-01158-f007:**
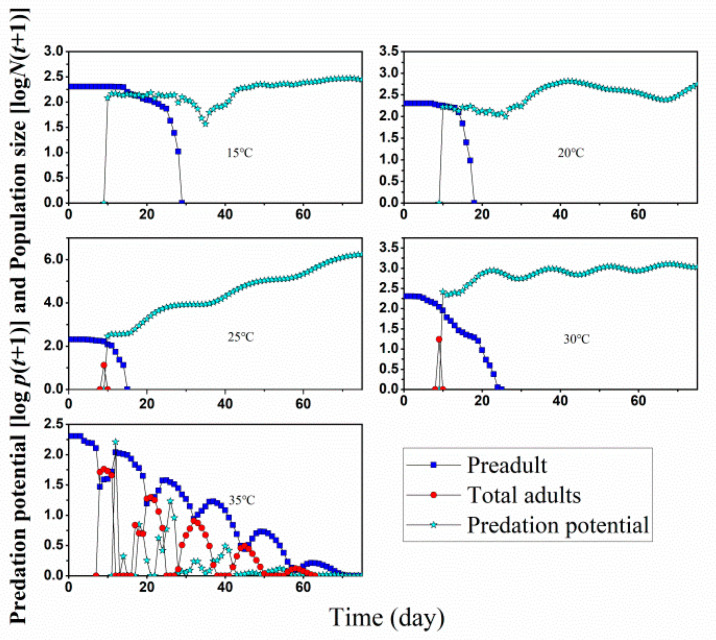
Simulated population growth of *D. minowai* starting form 200 eggs with five pairs of adults *O. minutus* released on the tenth day. Predation potential is the predation ability of *O. minutus* to *D. minowai* at time *t*.

**Table 1 insects-13-01158-t001:** Stages and developmental periods (days) (mean ± SE) of the *O. minutus* feeding on *D. minowai* at different temperatures.

Stage	15 °C		20 °C		25 °C		30 °C		35 °C	
	*n*	Mean ± SE	*n*	Mean ± SE	*n*	Mean ± SE	*n*	Mean ± SE	*n*	Mean ± SE
Egg	76	14.42 ± 0.06 a	66	8.24 ± 0.07 b	86	3.23 ± 0.07 c	69	2.03 ± 0.06 d	49	1.57 ± 0.07 e
1st nymphs	48	5.92 ± 0.09 a	52	4.67 ± 0.09 b	76	2.33 ± 0.06 d	53	2.58 ± 0.07 c	30	1.50 ± 0.09 e
2nd nymphs	31	6.13 ± 0.10 a	46	4.65 ± 0.08 b	66	3.12 ± 0.05 c	41	2.59 ± 0.08 c	20	2.70 ± 0.11 d
3rd nymphs	23	5.43 ± 0.11 a	33	4.52 ± 0.09 b	53	2.96 ± 0.05 c	29	1.79 ± 0.09 d	16	1.44 ± 0.13 e
4th nymphs	18	5.39 ± 0.16 a	22	4.00 ± 0.15 b	41	2.20 ± 0.09 c	20	1.10 ± 0.07 d	12	2.25 ± 0.18 c
5th nymphs	14	4.21 ± 0.11 a	16	4.06 ± 0.11 a	36	1.17 ± 0.06 c	14	1.00 ± 0 d	10	1.70 ± 0.15 b
Preadult	14	41.50 ± 0.20 a	16	30.38 ± 0.20 b	36	15.08 ± 0.12 c	14	11.07 ± 0.13 d	10	11.16 ± 0.37 d
Female adult longevity	9	24.11 ± 1.29 a	8	17.50 ± 1.39 b	23	16.13 ± 0.26 b	7	7.57 ± 0.66 c	6	3.67 ± 0.34 d
Male adult longevity	5	17.60 ± 3.26 a	8	12.75 ± 1.15 a	13	14.54 ± 1.14 a	7	6.14 ± 0.81 b	6	3.50 ± 0.29 c
Total longevity of females	9	65.56 ± 1.37 a	8	47.62 ± 1.40 b	23	31.13 ± 0.23 c	7	18.57 ± 0.72 d	6	14.17 ± 0.48 e
Total Longevity of males	5	59.20 ± 3.35 a	8	43.38 ± 1.18 b	13	29.77 ± 1.09 c	7	17.29 ± 0.87 d	4	15.25 ± 0.25 e
Adult preoviposition period (APOP)	9	9.67 ± 0.47 a	8	5.75 ± 0.53 b	23	1.87 ± 0.16 c	7	1.57 ± 0.43 c	6	2.00 ± 0.45 c
Total preoviposition period (TPOP)	9	51.11 ± 0.54 a	8	35.88 ± 0.48 b	23	16.87 ± 0.81 c	7	12.57 ± 0.48 d	6	12.5 ± 0.56 d
Oviposition days (*O_d_*)	9	6.22 ± 0.62 b	8	6.75 ± 0.65 b	23	10.00 ± 0.30 a	7	4.00 ± 0.49 c	6	2.00 ± 0.45 d

Note: Standard errors were estimated by using 100,000 resampling. Means followed by different letters are significantly different according to paired bootstrap test based on the confidence intervals of differences between treatments (*p <* 0.05). “*n*” represents the number of individuals survived to *j* stage.

**Table 2 insects-13-01158-t002:** Population statistics parameters (mean ± SE) of *O. minutus* starting from 100 eggs feeding on *D. minowai* at different temperatures.

Statistics Parameter	Mean ± SE				
	15 °C	20 °C	25 °C	30 °C	35 °C
Intrinsic rate of increase, *r* (d^−1^)	0.0066 ± 0.0065 b	0.0096 ± 0.0098 b	0.1224 ± 0.0089 a	0.0116 ± 0.0294 b	−0.1132 ± 0.00386 c
Finite rate of increase, *λ* (d^−1^)	1.0066 ± 0.0065 c	1.0096 ± 0.0098 c	1.1302 ± 0.0100 b	1.0117 ± 0.0291 c	0.8930 ± 0.0334 a
Net reproductive rate, *R*_0_ (offspring/individual)	1.45 ± 0.48 b	1.48 ± 0.52 b	14.76 ± 2.73 a	1.19 ± 0.45 c	0.21 ± 0.09 b
Mean generation time, *T* (d)	56.27 ± 1.16 a	40.83 ± 0.74 b	22.00 ± 0.14 c	15.00 ± 0.65 d	13.79 ± 0.52 d
Net predation rate, *c*_0_ (prey/individual)	74.54 ± 13.83 c	90.23 ± 13.27 b	310.92 ± 31.39 a	58.79 ± 8.01 d	17.29 ± 4.31 e
Finite predation rate, *ω* (d^−1^)	2.35 ± 0.28 d	4.11 ± 0.33 c	9.85 ± 0.48 a	7.41 ± 0.67 b	6.44 ± 0.76 b
Transformation rate, *Q_p_* (*c*_0_*/R*_0_)	55.50 ± 17.12 b	60.99 ± 31.05 b	21.07 ± 3.04 d	49.40 ± 33.42 c	82.33 ± 87.56 a
The gross reproduction rate (*GRR*)	12.05 ± 2.40 b	11.38 ± 2.76 b	42.23 ± 5.26 a	9.4 ± 2.68 c	2.42 ± 0.87 d
Fecundity (no. of eggs per female)	16.11 ± 1.53 b	18.50 ± 1.84 b	64.17 ± 1.53 a	17.00 ± 2.12 b	3.50 ± 0.67 c
F:M:N	9:5:86	8:8:84	23:13:64	7:7:86	6:4:90

Note: Based on paired bootstrap test with 100,000 resampling, the mean values of the columns following different letters were significantly different (*p <* 0.05). F stands for female and M for male, N represents the dead individuals which did not developed to adult stage.

**Table 3 insects-13-01158-t003:** Total predation rate (Mean ± SE) at each stage *j* excluding individuals that died in stage *j* (*P_j_*).

Stage	15 °C		20 °C		25 °C		30 °C		35 °C	
	*n*	Mean ± SE	*n*	Mean ± SE	*n*	Mean ± SE	*n*	Mean ± SE	*n*	Mean ± SE
1st nymphs	48	11.23 ± 0.44 h	52	12.48 ± 0.43 g	76	14.56 ± 0.57 f	53	10.39 ± 0.47 d	30	5.34 ± 0.45 e
2nd nymphs	31	22.55 ± 0.71 g	46	24.11 ± 1.10 f	66	44.64 ± 0.93 d	41	23.96 ± 1.10 b	20	14.75 ± 1.07 d
3rd nymphs	23	25.57 ± 1.27 f	33	41.48 ± 1.83 e	53	60.77 ± 1.84 c	29	24.55 ± 1.24 b	16	14.55 ± 1.42 d
4th nymphs	18	36.19 ± 2.05 e	22	50.50 ± 1.66 d	41	55.74 ± 2.45 c	20	18.65 ± 1.40 c	12	28.52 ± 2.40 c
5th nymphs	14	43.69 ± 2.23 d	16	49.00 ± 1.83 d	36	36.34 ± 2.21 e	14	19.79 ± 0.69 c	10	26.30 ± 2.32 c
Preadult	14	140.31 ± 3.06 c	16	183.68 ± 5.15 b	36	212.14 ± 3.94 b	14	97.86 ± 4.60 a	10	85.79 ± 5.15 a
Adult	14	262.07 ± 22.08 b	16	186.98 ± 15.11 b	36	503.71 ± 15.92 a	14	137.06 ± 11.03 a	10	52.58 ± 5.15 b
Female adult	9	303.12 ± 19.37 a	8	232.63 ± 14.97 a	22	527.60 ± 11.95 a	7	154.25 ± 13.13 a	6	47.01 ± 5.66 b
Male adult	5	188.25 ± 33.26 c	8	141.33 ± 13.18 c	14	466.11 ± 35.02 a	7	119.87 ± 16.53 a	4	60.76 ± 9.38 b

Note: In the paired bootstrap test with 100,000 resamplings, the mean values of the columns following different letters were significantly different (*p <* 0.05), and *P_j_* was the total predation rate that the predator successfully survived to the *j* stage and developed to the *j* + 1 stage (the sample size is *n_j_*). “*n*” represents the e number of individuals survived to *j* stage.

**Table 4 insects-13-01158-t004:** Predation rate at different stage including individuals that died at *j* (*U_j_*) of *O. minutus*.

Stage	15 °C		20 °C		25 °C		30 °C		35 °C	
	*m_j_*	Mean ± SE	*m_j_*	Mean ± SE	*m_j_*	Mean ± SE	*m_j_*	Mean ± SE	*m_j_*	Mean ± SE
1st nymphs	76	9.12 ± 0.47 h	66	11.38 ± 0.48 h	86	14.42 ± 0.55 f	69	9.74 ± 0.47 d	49	5.82 ± 0.39 d
2nd nymphs	48	17.73 ± 1.11 g	52	22.85 ± 1.10 g	76	43.29 ± 1.09 d	53	23.53 ± 1.04 c	30	14.57 ± 0.90 c
3rd nymphs	31	22.9 ± 1.37 f	46	36.69 ± 1.86 f	66	60.75 ± 2.08 c	41	24.49 ± 1.17 c	20	15.29 ± 1.26 c
4th nymphs	23	33.22 ± 2.27 e	33	43.75 ± 2.33 e	53	54.47 ± 2.23 c	29	20.11 ± 1.11 c	16	27.34 ± 2.02 c
5th nymphs	18	42.16 ± 1.92 d	22	43.76 ± 2.46 e	41	37.35 ± 2.01 e	20	22.70 ± 1.39 c	12	26.09 ± 1.99 b
Preadult	100	37.79 ± 5.15 d	100	60.33 ± 7.07 d	100	129.59 ± 8.83 b	100	39.61 ± 4.12 b	100	17.83 ± 2.99 b
Adult	14	261.85 ± 22.08 b	16	186.98 ± 15.11 b	36	503.71 ± 15.92 a	14	137.06 ± 11.03 a	10	52.58 ± 5.15 a
Female adult	9	303.12 ± 19.37 a	8	232.63 ± 14.97 a	22	527.60 ± 11.95 a	7	154.25 ± 13.13 a	6	47.01 ± 5.66 a
Male adult	5	188.2 ± 2.24 c	8	141.33 ± 13.18 c	14	466.11 ± 35.02 a	7	119.87 ± 16.53 a	4	60.76 ± 9.38 a

Note: In the paired bootstrap test of 100,000 resampling, the mean values of the columns following different letters were significantly different (*p <* 0.05), and *U_j_* was the individual who entered the *j* stage and developed to the *j* + 1 stage (the sample size is *m_j_*).

## Data Availability

The data presented in this study are available from the corresponding author with reasonable request.

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
