# Peer review of "Demographic Evaluation of the Control Potential of Orius minutus (Hemiptera: Anthocoridae) Preying on Dendrothrips minowai Priesner (Thysanoptera: Thripidae) at Different Temperatures"

_insects, 2022, doi:10.3390/insects13121158_

Round 1

Reviewer 1 Report

This article presents new and important information regarding the ability of Orius minutus to be used as a biological control agent against the tea thrips. Overall the paper is well-written, the design is sound and the statistical analyses are appropriate. The main issue with this manuscript is the English language which in many areas has grammatical errors which can lead to conclusion. A few specific errors regarding the results are given here: 

Lines 213-214 the sentence “At 15°C, the egg duration was 14.42 days, while it was only 1.57 days.” needs some clarification, seems to be missing something in regards to what was 1.57 days.

Lines 221-222 also need clarification “In addition, the oviposition time was the highest at 25℃, but only two days at 35°C”- what was the oviposition at 25 C?

Line 294-195 has a confusing sentence “The adult contribution to future populations was much greater than that of the worm at the five temperatures”- what is the worm? Is worm an error? Is it supposed to be nymph?

Lines 378-379 should read “western flower thrips”

Author Response

Dear reviewer, we are so appreciated your scrupulous review and great advice. We have taken care of all the revisions accordingly.

Question 1: Lines 213-214 the sentence “At 15°C, the egg duration was 14.42 days, while it was only 1.57 days.” needs some clarification, seems to be missing something in regards to what was 1.57 days.

Responses: Thanks for your valuable comments and suggestions. It has been modified to

"The egg duration was 14.42 days at 15°C but only 1.57 days at 35℃."

Question 2: Lines 221-222 also need clarification “In addition, the oviposition time was the highest at 25℃, but only two days at 35°C”- what was the oviposition at 25 C?

Responses: Thanks for your valuable comments and suggestions. It has been modified to “In addition, the oviposition time was the longest 10 days at 25°C dropped to the shortest 2 days at 35°C.”

Question 3: Line 294-195 has a confusing sentence “The adult contribution to future populations was much greater than that of the worm at the five temperatures”- what is the worm? Is worm an error? Is it supposed to be nymph?

Responses: Yes, and it has been modified as "nymph".

Question 4: Lines 378-379 should read “western flower thrips”

Responses: Done.

Reviewer 2 Report

The authors have graphed and presented their results clearly, drawing some attention to the implications of their findings. I found the study of interest and a good contribution to the knowledge of bioecology of biocontrol agents (BCAs). The methods used are appropriate for the objectives of the work and, in general, well depicted. The resulting figures are sufficient, informative, and of good quality helping to follow the reasoning throughout the manuscript.

The Intro and Discussion provide no insight on how this paper relates to the various other ones cited in the text or concerns that have been raised by other researchers. This article should provide details on all these fronts to provide the proper context for the work. Authors do not present any hypotheses or expectations that could be connected to previous studies; adding these details will improve the paper. The authors should clearly explain why the study was done, why it was important, and how it fits with the other studies. It should be clear and concise. The intro should also include what outcome(s) they expect, and how it would help support or refute their hypotheses or answer their questions.

My other concern is that the authors are extrapolating the applicability of their results beyond what the design supports. These are only data from a set of highly artificial constant temperatures, so the inference power of the paper is very limited, but authors do not acknowledge this detail at all and need to be more forthcoming. The possibility of using fluctuating temperature profiles was not investigated in this study. The interaction of cyclic temperatures with nonlinear characteristics of reproductive and predation parameters of BCAs can introduce significant deviations from the results obtained here, especially at the lower and higher temperatures of the reproductive activity functions. Therefore, studies across a broader set of fluctuating temperature regimes are still necessary to understand the real effect of temperature on reproductive and predation characteristics of BCAs, as this is the closest to the daily temperature fluctuations that occur in the field. So, I am suggesting to the authors to tone-down the language a little and admit that there are still substantive uncertainties to be considered.

Some of the authors statements would be much stronger if they tie their work to the body of literature that has built up on the bioecology and reproductive biology of other mass-produced endo- and ectoparasite biocontrol agents (BCAs) for field releases in California. They all point to the same direction and should be paired to this study. Some examples are J. Econ. Entomol. 112: 1560-1574 (mass produced ectoparasite BCAs) or J. Econ. Entomol. 112:1062-1072 (mass produced endoparasite BCAs), but there are others too. These studies provide strong evidence of increased longevity in BCAs reared at non-stressful low temperatures when compared to higher temperature regimes. Adding these details will improve the paper in my opinion. They further suggest that the parasitism or egg load was significantly higher at intermediate temperatures (20-30C) than at cline margins (<15C or >35C). This article should provide details on all these fronts to provide the proper context for the work. This is not to diminish the data gathered in this study, they are of value. But it is important for the authors not to overgeneralize, and to warn the reader, including regulatory agencies, against doing so as well. 

Good luck!

Author Response

Dear reviewer, we are so appreciated your scrupulous review and great advice. We have taken care of all the revisions accordingly.

Question 1: The authors have graphed and presented their results clearly, drawing some attention to the implications of their findings. I found the study of interest and a good contribution to the knowledge of bioecology of biocontrol agents (BCAs). The methods used are appropriate for the objectives of the work and, in general, well depicted. The resulting figures are sufficient, informative, and of good quality helping to follow the reasoning throughout the manuscript.

Response: Dear esteemed reviewer, thanks for your great comments and the following kind and precise information. We are very receptive to all your suggestion. And we do make sufficient adjustments accordingly.

The Intro and Discussion provide no insight on how this paper relates to the various other ones cited in the text or concerns that have been raised by other researchers. This article should provide details on all these fronts to provide the proper context for the work. Authors do not present any hypotheses or expectations that could be connected to previous studies; adding these details will improve the paper. The authors should clearly explain why the study was done, why it was important, and how it fits with the other studies.

Response: Thanks, and sure we agree with all these suggestions. We rewrote the introduction and discussion part concerning the necessity and importance of our research and its relationship with other reports.   

It should be clear and concise. The intro should also include what outcome(s) they expect, and how it would help support or refute their hypotheses or answer their questions.

Response: Yes, we added the description of questions and our expectations and outcomes.

My other concern is that the authors are extrapolating the applicability of their results beyond what the design supports.These are only data from a set of highly artificial constant temperatures, so the inference power of the paper is very limited, but authors do not acknowledge this detail at all and need to be more forthcoming.

Response: Thanks for your sincere critics. We realized that the early description was extrapolated, and we rewrote this related part and make the proper description.

The possibility of using fluctuating temperature profiles was not investigated in this study. The interaction of cyclic temperatures with nonlinear characteristics of reproductive and predation parameters of BCAs can introduce significant deviations from the results obtained here, especially at the lower and higher temperatures of the reproductive activity functions.Therefore, studies across a broader set of fluctuating temperature regimes are still necessary to understand the real effect of temperature on reproductive and predation characteristics of BCAs, as this is the closest to the daily temperature fluctuations that occur in the field.  So, I am suggesting to the authors to tone-down the language a little and admit that there are still substantive uncertainties to be considered.

Response: Yes, we agree that our result is of substantive uncertainties when considering the real effect of temperature. We moderate the description and make statements reasonable.

Some of the authors statements would be much stronger if they tie their work to the body of literature that has built up on the bioecology and reproductive biology of other mass-produced endo- and ectoparasite biocontrol agents (BCAs) for field releases in California.  They all point to the same direction and should be paired to this study. Some examples are J. Econ. Entomol. 112: 1560-1574 (mass produced ectoparasite BCAs) or J. Econ. Entomol. 112:1062-1072 (mass produced endoparasite BCAs), but there are others too. These studies provide strong evidence of increased longevity in BCAs reared at non-stressful low temperatures when compared to higher temperature regimes. Adding these details will improve the paper in my opinion. They further suggest that the parasitism or egg load was significantly higher at intermediate temperatures (20-30C) than at cline margins (<15C or >35C).

Response: Yes, we found that our result really fit in the linear model at a constant temperature profile as reported in J. Econ. Entomol. 112: 1560-1574 & J. Econ. Entomol. 112:1062-1072 for example longevity in O. minutus BCAs reared at low temperatures were increased when compared to higher temperature regimes. But we do not include how variation between the fluctuating ambient temperatures profile and constant temperatures. We referred both papers and tried illustrating this result in the discussion part.

This article should provide details on all these fronts to provide the proper context for the work. This is not to diminish the data gathered in this study, they are of value. But it is important for the authors not to overgeneralize, and to warn the reader, including regulatory agencies, against doing so as well. 

Thanks. we provide detail and illustrate proper context the in the revision.

And for your information, we made sufficient revisions(all marked) including English editing, some parts in the introduction and conclusion were completely reorganized and rewrote.

Reviewer 3 Report

Line 213 'The duration of egg stage was the longest among all stages. ' It is not always like what the authors described.

Lines 216-217, It is temperature-dependent, and no comparisons were performed. So, the results were not soundly described.

Lines 299-304 It is hard to understand why a reproductive value could be compared with the total preoviposition period.

Line 358 Discussion  In this part, most of sentences are redescribing the results. Furthermore, it is hard to understand the consistent between the conclusion and the references cited by the authors. For example, why reference 21 was cited in lines 379-381? And also, why did the authors cite reference 36 here (line 370)?

Author Response

Dear reviewer, we are so appreciated your scrupulous review and great advice. We have taken care of all the revisions accordingly.

Question 1: Line 213 'The duration of egg stage was the longest among all stages. ' It is not always like what the authors described.

Responses: Thanks. We have corrected this description. The duration of the egg stage appears to be the longest in preadult stages at lower temperatures 15-25℃.

Question 2:Lines 216-217, It is temperature-dependent, and no comparisons were performed. So, the results were not soundly described.

Responses: Done. We have corrected this confusing statement.

Question 3: Lines 299-304 It is hard to understand why a reproductive value could be compared with the total preoviposition period.

Responses: Thanks for your valuable comments and suggestions. We have corrected the puzzling sentence. It has been revised to “The results showed that the day of the highest reproductive value (vxj) was always close to the day of total preoviposition (TPOP). That is easy to understand that when TPOP is over, the insects started oviposition and reproductive value (vxj) arrived at peak value soon. For example, the highest reproduction value occurred on the 18th day at 25℃, which was close to the day of TPOP (16.87). The highest reproductive value appeared on the 50th day at 15℃, which was close to the days of TPOP (51.11). The highest reproduction value appeared on 8th day at 35℃, which was close to the day of TPOP(12.5)(Table 1, Figure 4B).”

Question 4: Line 358 Discussion In this part, most of sentences are redescribing the results. Furthermore, it is hard to understand the consistent between the conclusion and the references cited by the authors. For example, why reference 21 was cited in lines 379-381? And also, why did the authors cite reference 36 here (line 370)?

Responses: Thanks for your valuable comments and suggestions. Thank you for your valuable comments. We have modified the discussion part according to your comments and updated the references in this part. And References 21 and 36 really seemed not proper for context.

Round 2

Reviewer 2 Report

Authors have done a nice job addressing all of my original comments and those of other reviewers.  I have no further suggestions to improve the paper. Thank you!

Reviewer 3 Report

The authors have addressed the concerns. Some errors in formats need to be fixed. For example:  Lines 315, 367, 375, 388, and so on.